# Sex- and context-dependent effects of acute isolation on vocal and non-vocal social behaviors in mice

**Xin Zhao** [‡], **Patryk Ziobro** [‡], **Nicole M. Pranic** [‡], **Samantha Chu, Samantha Rabinovich, William Chan, Jennifer Zhao, Caroline Kornbrek, Zichen He, Katherine A. Tschida** *

Department of Psychology, Cornell University, Ithaca, NY, United States of America

‡ These authors share first authorship on this work.
* kat227@cornell.edu

**Data Availability Statement:** All data associated with this study are made available through Cornell eCommons (https://doi.org/10.7298/z4x4-0160).

**Funding:** This research was supported by a research grant from the Cornell Center for Social

## Abstract

Humans are extraordinarily social, and social isolation has profound effects on our behavior, ranging from increased social motivation following short periods of social isolation to increased anti-social behaviors following long-term social isolation. Mice are frequently used as a model to understand how social isolation impacts the brain and behavior. While the effects of chronic social isolation on mouse social behavior have been well studied, much less is known about how acute isolation impacts mouse social behavior and whether these effects vary according to the sex of the mouse and the behavioral context of the social encounter. To address these questions, we characterized the effects of acute (3-day) social isolation on the vocal and non-vocal social behaviors of male and female mice during same-sex and opposite-sex social interactions. Our experiments uncovered pronounced effects of acute isolation on social interactions between female mice, while revealing more subtle effects on the social behaviors of male mice during same-sex and opposite-sex interactions. Our findings advance the study of same-sex interactions between female mice as an attractive paradigm to investigate neural mechanisms through which acute isolation enhances social motivation and promotes social behavior.

## Introduction

Social interactions form the backbone of our experiences as humans. We find social interactions intrinsically rewarding, and we are highly motivated to seek out social contact and to form and maintain social bonds. Consequently, the experience of social isolation is aversive and increases our motivation to seek out and engage in social interactions [1–3]. Rodents are frequently used as a model to understand how social isolation impacts emotional states and engagement in social behaviors in humans. As in humans, chronic social isolation in rodents leads to increases in anti-social behaviors, including increased anxiety, increased aggression, and decreased social motivation [4–14]. Many studies have also reported effects (or lack thereof) of chronic social isolation on the production of ultrasonic vocalizations (USVs) [15–19], which many rodents emit during same-sex and opposite-sex social encounters [20, 21].

Sciences (awarded to K.A.T.). https://socialsciences.cornell.edu/ The funders had no role in study design, data collection and analysis, decision to publish, or preparation of the manuscript.

**Competing interests:** The authors have declared that no competing interests exist.

However, much less is known about how rodent social behavior is affected by short-term social isolation. A small number of studies have reported that acute social isolation promotes subsequent social interaction in rodents [22–25], but it remains unknown whether the production of social USVs is similarly enhanced during social encounters following acute isolation. Furthermore, most studies to date have characterized the effects of isolation on social behavior in a single context, often focusing on social interactions between male rodents [15, 17, 22, 23]. Thus, it remains unknown whether the effects of acute isolation on rodent social behavior are generalized across many types of social interactions, or alternatively, vary according to sex and social context.

To address these questions, we examined the effects of acute isolation on social behavior in mice, because they offer an extensive genetic toolkit that will facilitate future investigation of molecular and circuit mechanisms underlying the effects of isolation on social behavior. Specifically, we examined the effects of acute (3-day) social isolation on the vocal and non-vocal social behaviors of adult male and female B57BL/6J mice during subsequent same-sex and opposite-sex interactions. We found that acute isolation exerts both sex- and context-dependent effects on USV production and non-vocal social behaviors, with particularly strong effects on the vocal and non-vocal social behaviors of same-sex pairs of females. Furthermore, we found differences in how USV production is related to non-vocal social behaviors in different social contexts, revealing sex- and context-dependent differences in the coupling between USV production and social motivation. This study provides the first direct comparison of the effects of acute isolation on mouse social behavior across sex and social context, and our findings indicate that same-sex interactions between female mice are an attractive paradigm to investigate the neural mechanisms through which acute isolation enhances social motivation and promotes social interaction.

## Materials and methods

Further information and requests for resources and reagents should be directed to the corresponding author, Katherine Tschida (kat227@cornell.edu).

### Ethics statement

All experiments and procedures were conducted according to protocols approved by the Cornell University Institutional Animal Care and Use Committee (protocol #2020–001).

### Subjects

Male and female C57BL/6J mice (Jackson Laboratories, 000664) were housed with their same-sex siblings following weaning until the beginning of the experiment (>7 weeks of age). Mice were kept on a 12:12 reverse day-night cycle and given unlimited access to water and chow. The estrous state of female mice was not monitored.

### Behavioral experiments

We used a between-subjects design to measure the effects of social isolation on vocal and non-vocal social behaviors in different social contexts. Specifically, three days prior to the day of behavioral measurements, male and female subject mice either continued residing with their same-sex siblings (group-housed residents) or were single-housed until the day of recording (single-housed residents). On the day of the behavior measurements, the subject mouse was transferred in its home cage to a recording chamber (Med Associates) equipped with an ultrasonic microphone (Avisoft), infrared light source (Tendelux), and webcam (Logitech, with

infrared filter removed to enable video recording under infrared lighting). The home cage was either placed inside a custom acrylic chamber (similar dimensions to the home cage but taller) or was fitted with a custom lid to permit USV recordings. In the case of group-housed residents, the mouse's siblings were removed from the home cage and transferred to a clean cage for the duration of the test. A novel, unfamiliar group-housed visitor mouse (male or female depending on the context) was then placed in the subject animal's home cage for 30 minutes, and video and audio recordings were made. Visitor mice were used across multiple experiments, including those with single-housed residents and those with group-housed residents, and were therefore socially experienced. Many of our subject animals were socially naïve at the time of the experiment (i.e., no social experience other than with their same-sex siblings), and a subset of the subject animals (21/53 female residents in female-female recordings, 26/41 residents in male-male-recordings) were previously given brief (~5 minute) social experiences with novel female conspecifics prior to postnatal day 40 as part of another study (no greater than 40 minutes in total). Resident animals that had this limited amount of prior social experience were counterbalanced between group-housed and single-housed conditions. USV rates were not significantly different between pairs that contained naïve residents versus pairs that contained residents with brief social experiences (p = 0.08 for difference between naïve and experienced in female-female trials; p = 0.28 for difference in male-male trials, Mann Whitney U test). With the exception of these animals that had brief prior social experiences, all other resident animals (including all male residents used in the male-female context) were socially and sexually naïve prior to the test.

Visitors were marked with acrylic paint or hair dye to facilitate identification in videos. We made measurements of vocal and non-vocal behaviors in three contexts: female residents with female visitors, male residents with male visitors, and male residents with female visitors.

## Behavior analysis

Trained observers scored the following categories of behavior from webcam videos: (1) resident and visitor not interacting, (2) resident-initiated social interaction (sniffing, following, or chasing), (3) visitor-initiated social interaction, (4) resident mounting the visitor, (5) visitor mounting the resident, (6) resident-initiated fighting (i.e., attacking and/or biting), (7) visitor-initiated fighting, and (8) mutual social interaction (mutual sniffing).

## USV recording and analysis

USVs were recorded using an ultrasonic microphone (Avisoft, CMPA/CM16), connected to an Avisoft recording system (UltrasoundGate 166H, 250 kHz sample rate). In pilot experiments, USVs were detected using codes modified from those provided by the Holy lab (http://holylab.wustl.edu/) using the following parameters (mean frequency > 45 kHz; spectral purity > 0.3; spectral discontinuity < 0.85; min. USV duration = 5 ms; minimum inter-syllable interval = 30 ms). We found, however, that some USVs were low amplitude and not detected accurately by our automated codes, particularly USVs emitted during male-male interactions. To ensure the highest accuracy of USV detection, trained observers manually annotated USVs from wav files using custom Matlab codes.

## Quantification and statistical analyses

To determine whether to use parametric or non-parametric statistical tests for a given comparison, we examined the normality of the residuals for the relevant data distributions (determined by visual inspection of plots of z-scored residuals). In cases in which the residuals for one group's data were not normally distributed, we employed a non-parametric statistical test.

In cases in which parallel comparisons were made for all three social contexts (male-male, female-female, and male-female) and in which the residuals for one's group data were non-normally distributed, we opted to employ non-parametric statistical tests for all parallel comparisons for consistency and to be conservative in our conclusions. For both parametric and non-parametric comparisons, two-sided statistical comparisons were used (alpha = 0.05). Details of the statistical analyses used in this study are included in S1 Table in S1 File. No statistical methods were used to predetermine sample sizes. Error bars represent standard deviation unless otherwise noted. Violin plots were created using code from Holger Hoffmann (2021, Matlab Central File Exchange, https://www.mathworks.com/matlabcentral/fileexchange/45134-violin-plot).

## Results

To measure the effects of acute social isolation on the vocal and non-vocal social behaviors of mice, we performed video and audio recordings during 30-minute social encounters between opposite-sex and same-sex pairs of mice (i.e., female-female, male-male, and male-female interactions). Recordings were performed in the home cage of the resident animal, which had been either group-housed continuously with its same-sex siblings or single-housed for three days prior to the day of the experiment. An unfamiliar group-housed visitor mouse was then introduced to the resident's home cage, and vocal and non-vocal social behaviors were recorded. In the case of male-female encounters, the resident mouse was always the male. During interactions between same-sex pairs of mice, previous studies have found that either mouse in the pair can produce ultrasonic vocalizations (USVs) [26, 27]. For this reason, we made no assumptions about which mouse in a same-sex pair was vocalizing at a given time and report the total number of USVs produced by the pair. In the case of opposite-sex interactions, previous work has shown that males produce the majority (~85%) of USVs during interactions with females [28, 29], and we similarly assume that most USVs are produced by the resident male mouse. We examined the effects of acute social isolation on USV production, non-vocal social behaviors, and the relationship between vocalizations and non-vocal social behaviors in these three social contexts.

### Effects of acute isolation on USV production in same-sex and opposite-sex interactions

We first measured the effects of acute social isolation on the production of USVs during interactions between pairs of females. We observed robust effects of acute social isolation on the vocal behavior of females, and the total number of USVs produced in female-female interactions was nearly four times higher in pairs with a single-housed resident compared to pairs with a group-housed resident (Fig 1A; 1441 ± 654 USVs in N = 22 single-housed resident trials vs. 369 ± 357 USVs in N = 31 group-housed resident trials; p < 0.0001, Mann Whitney U test). USV production tended to peak during the first 5 minutes in both types of pairs, although female-female pairs with a single-housed resident emitted USVs with a shorter average latency (Fig 1A, S1 Fig in S1 File; mean latency = 13.1 s for single-housed resident trials vs. 89.0 s for group-housed resident trials, p = 0.03, Mann Whitney U test). The temporal distribution of USVs produced throughout the 30-minute trial also differed between these groups, and female-female pairs with a group-housed resident produced a larger proportion of their total USVs in the first 5 minutes (5 min USVs/total USVs = 0.62 ± 0.35 for group-housed resident trials vs. 0.40 ± 0.17 for single-housed resident trials, p = 0.02, Mann Whitney U test). We conclude that acute social isolation robustly enhances the production of USVs during social interactions between pairs of females.

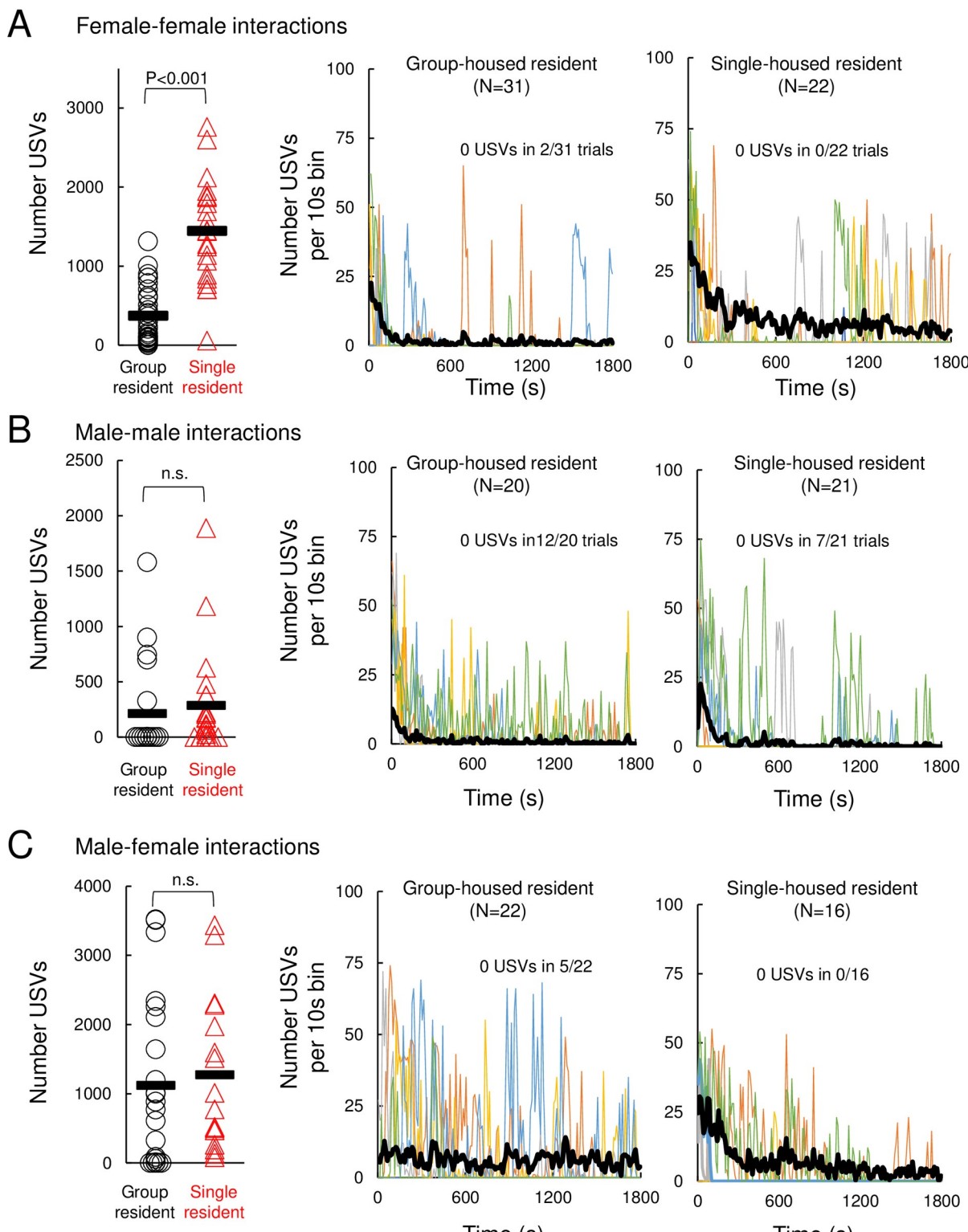

**Fig 1. Effects of acute isolation on USV production during same-sex and opposite-sex social encounters.** (A) Number of USVs recorded is shown for female-female social encounters, in which a group-housed female visitor was introduced into the home cage of either a group-housed or a single-housed resident. Left panel shows total number of USVs produced during social encounters with group-housed and single-housed female residents. Middle panel (trials with group-housed residents) and right panel (trials with single-housed residents) show the dynamics of USV production over time, plotted as total number of USVs produced in each 10s-long bin. Black line shows mean values for

each condition, and thinner colored lines show 5 representative trials. (B) Same as (A), for male-male social encounters. (C) Same as (A), for male-female social encounters. Please note the difference in y axis ranges for the left-most plots in (A)-(C).

In contrast to the strong effects of isolation on female-female vocal behavior, there was no significant effect of acute isolation on the total number of USVs emitted during male-male encounters or during male-female encounters (Fig 1B and 1C; p = 0.15 for N = 20 group-housed resident vs. N = 21 single-housed resident male-male trials; p = 0.44 for N = 22 group-housed resident vs. N = 16 single-housed resident male-female trials, Mann Whitney U tests). Although acute isolation had no significant effect on the total number of USVs recorded during male-male and male-female interactions, we noted subtle effects of isolation on male vocal behavior. First, out of the 20 male-male trials with group-housed residents, USVs were detected in 8 trials. However, we noted that 3 of these 8 trials had only very low rates of USV production (1–3 USVs detected over the 30-minute trial). If we considered only male-male trials in which moderate rates of USVs were recorded (>25 USVs), we found that a significantly greater proportion of male-male trials with a single-housed resident had high USV production (14 of 21 trials with a single-housed resident vs. 5 of 20 trials with a group-housed resident; p = 0.02, z-test for two independent proportions). Male-male pairs with a single-housed resident also emitted a higher proportion of their total USVs in the first 5 minutes of the trial than pairs with a group-housed resident (5 min USVs/total USVs = 0.84 ± 0.19 for single-housed resident trials vs. 0.51 ± 0.38 for group-housed resident trials, p = 0.04, Mann Whitney U test). Second, in male-female interactions, we noted that USVs were emitted earlier in trials with a single-housed male resident than in trials with a group-housed resident) (Fig 1C, S1 Fig in S1 File; mean latency = 35.3 s for single-housed resident trials vs. 182.9 s for group-housed resident trials; p = 0.03, Mann Whitney U test; p = 0.61 for difference in USV latency in male-male trials). Finally, the temporal distribution of USVs differed between groups, and male-female pairs with a single-housed resident emitted a higher proportion of their total USVs in the first 5 minutes of the trial than pairs with a group-housed resident (5 min USVs/total USVs = 0.52 ± 0.27 for single-housed resident trials vs. 0.27 ± 0.31 for group-housed resident trials, p = 0.01, Mann Whitney U test). In summary, acute social isolation enhances USV production between pairs of female mice and exerts more subtle effects on USV production by males during same-sex and opposite-sex interactions.

### Effects of acute isolation on non-vocal social behaviors in same-sex and opposite-sex interactions

Acute isolation exerts sex- and context-dependent effects on USV production, and we wondered whether isolation also impacted non-vocal social behaviors differentially in these groups of mice. To examine the effects of isolation on the overall amount of social interaction, we considered non-vocal social behavior as the sum of all types of social interactions observed in a trial (i.e., sniffing, following, chasing, mounting, and fighting). We found that pairs of females with a single-housed resident spent significantly more time engaged in social interaction than pairs with a group-housed resident (Fig 2A; pairs with single-housed resident spent 21.8 ± 10.5% of trial time interacting vs. 7.3 ± 4.1% in pairs with group-housed residents; p < 0.001, Mann Whitney U test). In male-male interactions, acute social isolation of the resident mouse tended to increase the time that males spent interacting, although this effect was not significant (Fig 2A; pairs with single-housed resident spent 14.5 ± 9.2% of trial time interacting vs. 9.2 ± 4.9% in pairs with group-housed residents p = 0.08, Mann Whitney U test). Similarly, in male-female interactions, acute social isolation of the resident male tended to

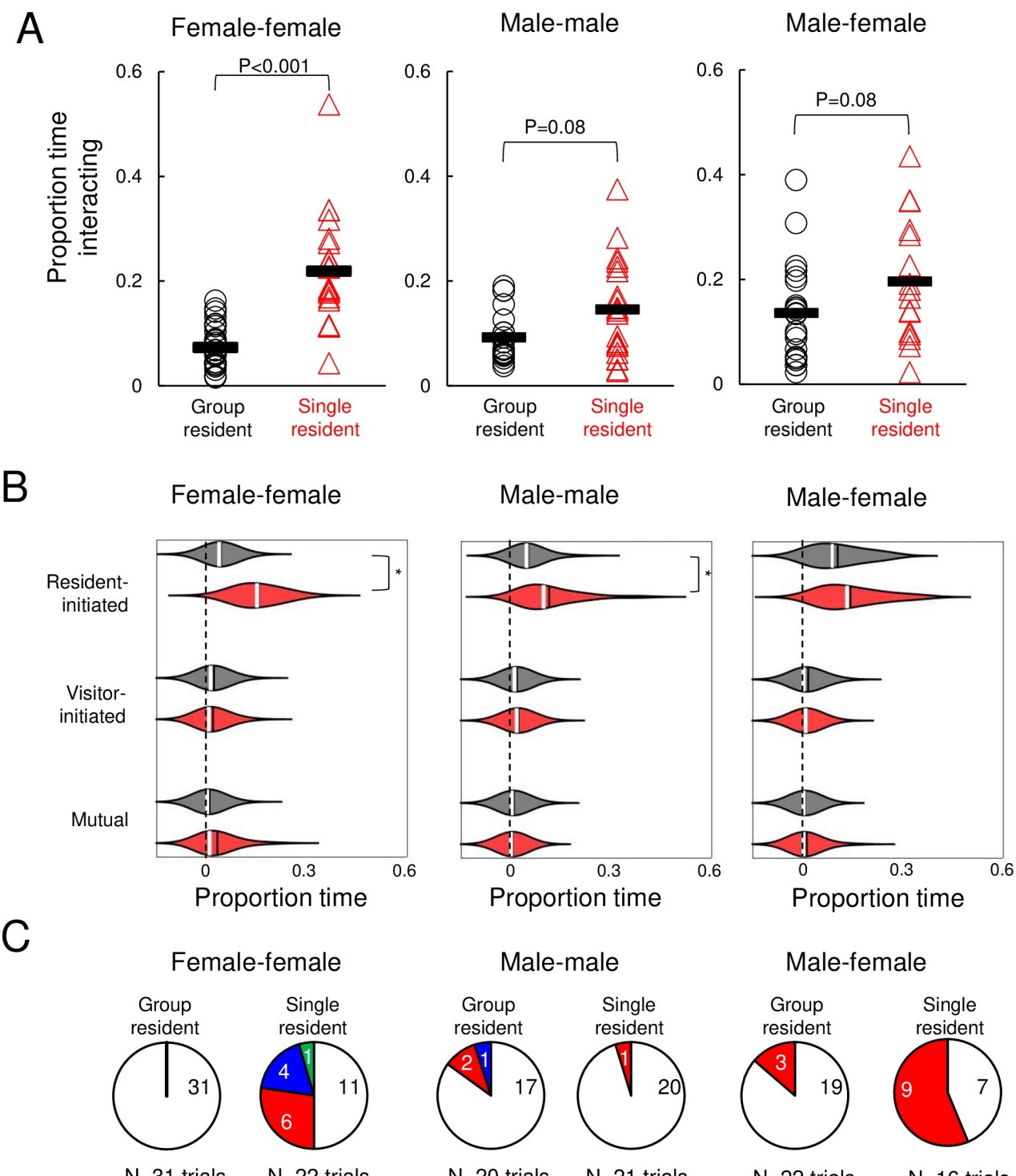

**Fig 2. Effects of acute isolation on non-vocal social behaviors.** (A) Proportion time spent engaged in social interaction is shown for female-female trials (left), male-male trials (middle), and male-female trials (right). (B) Violin plots show the proportion time spent engaged in different non-vocal social behaviors for female-female trials (left), male-male trials (middle), and male-female trials (right). Gray, trials with group-housed residents; red, trials with single-housed residents. White lines indicate median values, black lines indicate mean values. Asterisks, p < 0.05. (C) Pie charts show the number of trials with mounting in female-female trials (left), male-male trials (middle), and male-female trials (right). White, no mounting; red, resident-initiated mounting; blue, visitor-initiated mounting; green, trials with both resident- and visitor-initiated mounting.

increase the time the pair spent interacting, but these effects were not significant (Fig 2A; pairs with single-housed resident spent 19.6 ± 11.8% of trial time interacting vs. 13.6 ± 9.3% in pairs with group-housed residents p = 0.08, Mann Whitney U test). In summary, acute social isolation tends to increase the amount of time spent interacting in both same-sex and opposite-sex pairs, although it does so most robustly (and at the level of statistical significance) for female-female pairs.

We next considered the effects of acute social isolation on specific types of non-vocal social behaviors, by categorizing periods of social interaction as resident-initiated social interaction (i.e., sniffing, following, chasing), visitor-initiated social interaction, and mutual social interaction (i.e., mutual sniffing). Although mounting and fighting were observed in a subset of trials, these behaviors occupied a small percentage of the total trial time (typically <5%) and are considered separately below. In female-female trials, we found that single-housed residents spent significantly more time initiating social interactions than group-housed residents (sniffing, following, and chasing), and there were no significant differences in visitor-initiated social interaction or mutual interaction (Fig 2B, left; two-way ANOVA with repeated measures on one-factor followed by post-hoc tests; p < 0.001 for difference in resident-initiated interaction). Similarly, in male-male trials, single-housed residents spent significantly more time initiating social interaction than group-housed residents (Fig 2B, middle; p = 0.003 for difference in resident-initiated interaction). Finally, in male-female trials, there were no significant differences in the proportion of time spent in resident-initiated, visitor-initiated, or mutual social interaction (p > 0.05). Thus, acute social isolation exerts sex- and context-dependent effects on social interaction, increasing resident-initiated social interaction in same-sex pairs of males and females but not in opposite-sex pairs of mice.

We next measured the effects of acute isolation on mounting behavior. Male mice commonly mount females during opposite-sex interactions, and mounting has also been reported to a lesser extent during same-sex interactions between males and females [30–33]. Surprisingly, we observed a robust effect of acute isolation on mounting in female-female pairs. We never observed mounting in female-female pairs with group-housed residents (Fig 2C, 0/31 trials). In contrast, mounting was recorded in 11 of 22 trials with single-housed residents (Fig 2C, p < 0.0001 for difference between groups, z-test for two independent proportions). Within these 11 trials, we observed mounting events initiated by residents and by visitors (N = 6 trials with resident-initiated mounting, N = 4 trials with visitor-initiated mounting, and N = 1 trial with mounting initiated by both resident and visitor). In male-male trials, mounting was observed infrequently, and there was no apparent effect of acute social isolation on mounting (Fig 2C, observed in 3 of 20 trials with group-housed residents and in 1 of 21 trials with single-housed residents, p > 0.05). Finally, we found that acute social isolation increased mounting behavior of males during interactions with females (mounting in 3 of 22 male-female trials with group-housed residents vs. 9 of 16 trials with single-housed residents, p = 0.005). In summary, acute social isolation promotes mounting between same-sex pairs of females and in opposite-sex pairs of mice.

Finally, we examined the effects of acute isolation on fighting, given that chronic social isolation is known to increase aggression in male rodents (4, 7, 10, 11, 13, 14). We never observed fighting in female-female and male-female trials (0/53 female-female trials, 0/38 male-female trials). In male-male trials, fighting was observed only infrequently, and there was no apparent effect of acute social isolation (fighting observed in 2 of 20 trials with group-housed residents and in 4 of 21 trials with single-housed residents, p = 0.41, z-test for two independent proportions). We conclude that acute social isolation does not significantly affect levels of aggression in males and female B57BL/6J mice.

## Effects of acute isolation on the relationship of USV production to non-vocal social behaviors

We next characterized the relationship between USV production and non-vocal social behaviors during same-sex and opposite-sex interactions and asked whether acute isolation affects the relationship between vocalization and other social behaviors. In short, what are mice doing when they vocalize, and is that relationship affected by acute isolation? We examined the relationship between USV production and non-vocal social behaviors in general by comparing the total number of USVs produced to the total time spent interacting in the three different social contexts (Fig 3A). In female-female pairs with a group-housed resident, we observed a significant positive relationship between the number of USVs produced and the proportion of time in the trial the mice spent interacting (Fig 3A, left, black symbols, p = 0.002 for linear regression, $R^2 = 0.28$). Female pairs with a single-housed resident spent more time interacting and produced more USVs, but we again observed a significant positive relationship between USV production and time spent interacting (Fig 3A, left, red symbols, p = 0.001 for linear regression, $R^2 = 0.42$). In summary, USV production is correlated with social interaction time in female pairs, and acute isolation potentiates social interaction during female-female encounters and drives a concomitant increase in USV production.

In contrast to our observations in female-female pairs, USV production was not well related to the total time spent interacting in male-male pairs with a group-housed resident (Fig 3A, middle, black symbols, p = 0.83 for linear regression). Notably, this relationship was altered by acute isolation, and in male-male pairs with a single-housed resident, USV production was positively related to total time spent interacting (Fig 3A, middle, red symbols, p = 0.02, $R^2 = 0.39$). Finally, USV production in male-female interactions was significant correlated with the total time the mice spent interacting, and this was true for trials with group-housed residents as well as trials with single-housed residents (p = 0.003, $R^2 = 0.36$ for linear regression in group-housed resident trials, p = 0.005, $R^2 = 0.43$ for single-housed resident trials). We conclude that acute isolation drives the emergence of coupling between USV production and non-vocal social behaviors in male-male interactions and has no effect on the relationship between USV production and total social interaction time in male-female interactions.

What proportion of USVs are produced during different non-vocal social behaviors in same-sex and opposite-sex interactions, and does acute isolation impact these relationships? To visualize the relationship between vocal and non-vocal behaviors in each trial, we created ethograms in which USV rates are plotted over time against the production of different categories of non-vocal social behavior for each trial in our dataset (S2-S7 Figs in S1 File). We then calculated the proportion of the total USVs in each trial produced during different types of non-vocal social behavior in each of the 6 groups of mice (trials with <25 USVs were excluded from this analysis). We first considered the relationship of USV production to resident-initiated, visitor-initiated, and mutual social interactions, and we separately consider the relationship of USV production to mounting and fighting below.

Female-female pairs with a single-housed resident produced more USVs during resident-initiated social interactions than pairs with a group-housed resident, and there were no significant differences in the proportion of total USVs produced during visitor-initiated and mutual interactions (Fig 3B, two-way ANOVA with repeated measures on one-factor, followed by post-hoc tests; p = 0.03 for difference in proportion of USVs produced during resident-initiated social interactions). We noted previously that resident-initiated interactions were significantly increased in female-female pairs with a single-housed resident, consistent with the idea that acute isolation increases USV production concomitantly with the increase in resident-initiated interactions (compare Fig 3B left to Fig 2B left). Similarly, male-male pairs with a single-

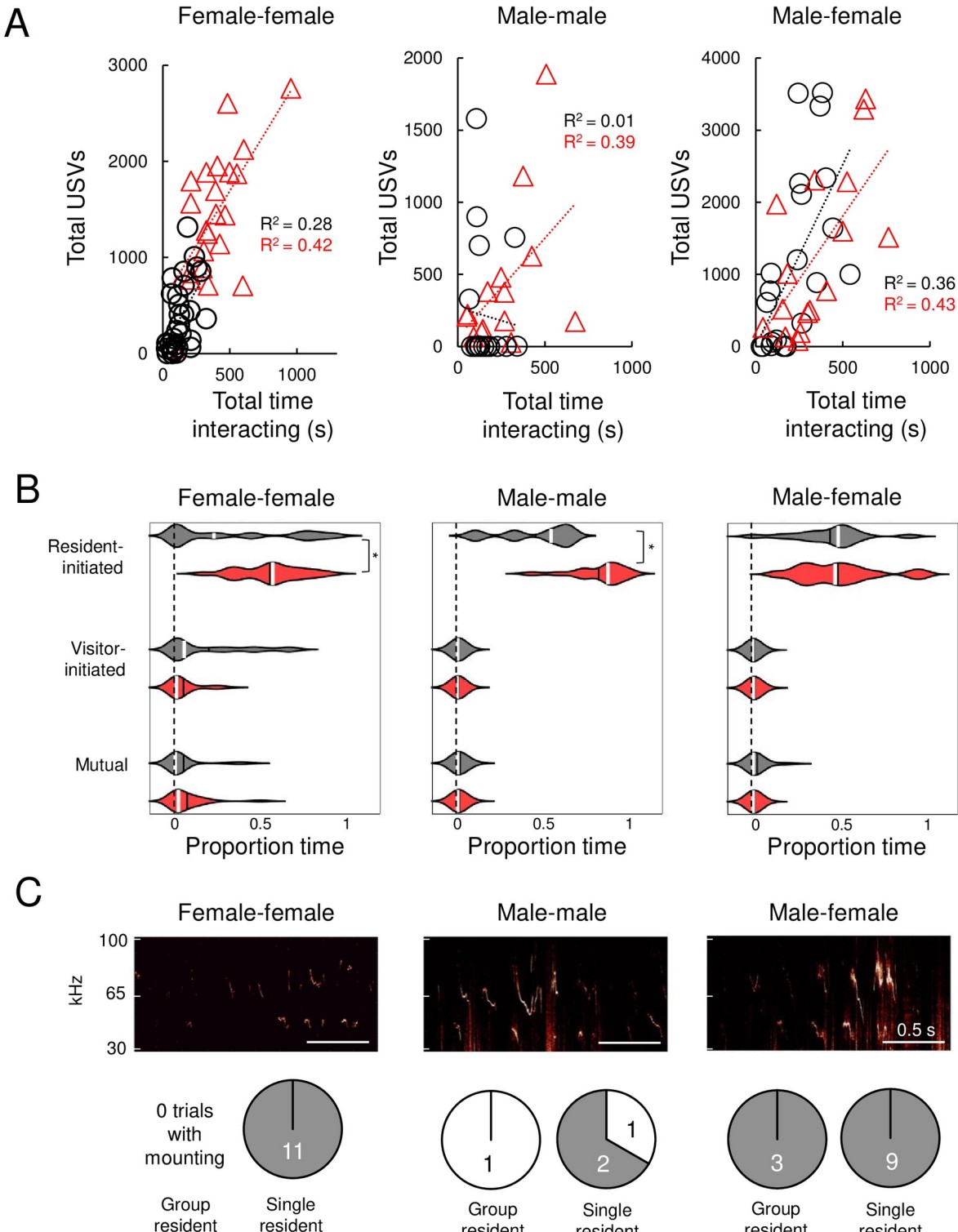

**Fig 3. Effects of acute isolation on the relationship of USV production to non-vocal social behaviors.** (A) Scatterplots show the number of USVs emitted versus the proportion total time spent interacting for female-female trials (left), male-male trials (middle), and male-female trials (right). Black symbols, trials with group-housed residents; red symbols, trials with single-housed residents. (B) Violin plots show the proportion of USVs produced during different types of behavior. Gray plots summarize data for trials with group-housed residents, red plots summarize data for trials with single-housed residents. White lines indicate median values, black lines indicate mean values. Asterisks, p < 0.05. (C) Spectrograms show USVs produced during mounting events during female-female (left), male-male

(middle), and male-female (right) social interactions. Pie charts show the number of trials in which mounting was accompanied by USV production (gray) versus trials in which mounting was not accompanied by USV production (white) during female-female (left), male-male (middle), and male-female (right) social interactions.

housed resident produced a significantly higher proportion of total USVs during resident-initiated social interactions, in line with the observed increase in resident-initiated social interactions in male-male pairs with a single-housed resident (compare Fig 3B middle to Fig 2B middle). Finally, there were no significant differences in the proportion of total USVs produced during these three types of social behavior in male-female interactions with a group-housed vs. single-housed resident (Fig 3B, right).

We then examined the relationship of USV production to mounting in the six groups of animals. As expected from previous work [34, 35], we found that mounting by males during opposite-sex social interactions was accompanied by USV production (Fig 3C, right). Surprisingly, we found that all mounting events during same-sex female interactions were accompanied by USV production as well (Fig 3C, mounting was accompanied by USV production in 11 of 11 trials with mounting). Finally, during male-male interactions, mounting was accompanied by USV production in 2 of the 4 total cases (Fig 2C). In the two trials in which male-male mounting events were not accompanied by USV production, we note that mounting events immediately preceded fights (no fights were observed in the male-male trials in which mounting was accompanied by USV production). Because we only observed a small number of fights in our male-male trials, we cannot make any conclusions regarding the relationship between USV production and subsequent fighting. We note, however, that USV production never occurred during fights (S2-S7 Figs in S1 File). In summary, acute isolation promoted mounting during female-female and male-male social interactions, and these mounting events were accompanied by USV production.

## Discussion

In this study, we measured the effects of acute social isolation on vocal and non-vocal social behaviors of mice during same-sex and opposite-sex interactions. Acute social isolation had a profound effect on social interactions between pairs of females. Female pairs with a single-housed resident produced more USVs and spent more time interacting. In particular, single-housed female residents spent more time initiating social interactions with visitors, and we also observed the emergence of mounting accompanied by USV production in female pairs with single-housed residents, which was never observed in trials with group-housed residents. In contrast to these robust effects on same-sex interactions in females, acute isolation had more subtle effects on social interactions between males. Although acute isolation did not significantly affect the total number of USVs produced during male-male interactions, a greater proportion of male pairs with a single-housed resident produced high rates of USVs (>25 USVs). Single-housed resident males spent more time initiating social interactions with visitors than group-housed residents, and acute isolation increased the coupling between USV production and non-vocal social interactions in male-male pairs. Finally, in male-female pairs, there were no effects of acute isolation of the resident male on the total number of USVs produced or the time spent interacting, although single-housed resident males were more likely to mount females than group-housed residents. We conclude that the effects of acute isolation on vocal and non-vocal social behavior vary according to sex and to social context, with the greatest impacts observed on interactions between pairs of females.

Why does acute isolation more robustly affect social interactions between females than interactions between males or opposite-sex pairs of mice? One idea is that interactions

between pairs of females are affiliative and thus are strongly influenced by levels of pro-social motivation. Indeed, affiliative interactions between female mice are common, with demes of wild house mice frequently including several breeding and several non-breeding females [36, 37]. In addition, female mice sometimes engage in communal nesting and nursing [38]. In further support of this idea, social interactions between female mice are associated with neuronal activation within mesolimbic reward circuits, and artificial activation of dopaminergic inputs to the nucleus accumbens promotes social interaction between females [39]. Acute social isolation may enhance pro-social motivation in females and thereby increase the production of vocal and non-vocal social behaviors in subsequent interactions between pairs of females. In contrast, interactions between males and females are dominated by male-initiated behaviors and likely reflect levels of sexual motivation rather than a more generic form of affiliative social motivation. Although we found that acute isolation increased male mounting during interactions with females, the lack of effects on other aspects of vocal and non-vocal behavior during male-female encounters suggests that acute isolation does not strongly impact male sexual motivation. We note, however, that the male residents used in our male-female interactions were all sexually naïve, and it is possible that acute isolation could have more pronounced or different effects on the vocal and non-vocal behaviors of sexually experienced males. Finally, interactions between unfamiliar pairs of male mice are likely more agonistic than affiliative and thus are not as strongly impacted by acute social isolation as interactions between pairs of females. Although we observed low rates of fighting during the 30 minute-long male-male encounters used in our study, unfamiliar male mice engage in agonistic behaviors including chasing, mounting, and fighting to establish social hierarchies when housed together for longer periods of time [32, 40]. In summary, one possibility is that acute social isolation strongly promotes affiliative social motivation and thereby strongly influences female-female social encounters, while exerting less pronounced effects on sexual and aggressive motivation during the social interactions of male mice.

We note, however, that the interpretation that isolation-induced increases in female social interaction are affiliative is somewhat complicated by our finding that acute isolation increases the occurrence of mounting during female-female interactions. Same-sex mounting in female mice has been described [31], and although its behavioral functions remain unclear, there is evidence that females use mounting to establish social dominance over other females [33]. Of particular interest is our finding that female-female mounting is accompanied by USV production. Same-sex mounting between pairs of male mice is considered to be a low-level aggressive behavior [32], is controlled by a hypothalamic brain region important for aggression [30], and is typically not accompanied by USV production [30]. This contrasts with male mounting of female mice, which is controlled by a hypothalamic brain region important for male sexual behavior [30, 41–44] and is typically accompanied by USV production (Fig 3C) [30, 34, 35]. At present, the neural circuits that regulate female-female mounting remain unknown, and the elucidation of these circuits in future studies may shed light on the function of same-sex mounting between pairs of female mice, as well as on whether the isolation-induced increases in female social interaction observed in the current study reflect an increase in affiliative or aggressive motivation.

What neural mechanisms underlie the potentiating effects of acute isolation on social interactions between pairs of females? Previous work has shown that chronic social isolation can exert sex-specific effects on female neural circuits and particularly on neuroendocrine signaling. Social isolation from the time of weaning alters the intrinsic properties of neurons in the paraventricular nucleus of the hypothalamus (PVN) that express corticotrophin-releasing hormone in female but not male mice [45]. In prairie voles, a 4-week social isolation elevates the density of oxytocin-expressing neurons in the PVN and elevates plasma levels of oxytocin in

females but not in males [46]. In addition, although it remains unclear whether the isolation-induced increases in female interaction that we observed are affiliative or aggressive in nature, we note that female aggression is regulated by estradiol [47] and that social hierarchy position in female mice is associated with changes in estrogen receptor expression within the ventromedial hypothalamus [33]. Future studies can examine whether changes in neuroendocrine and/or hormonal signaling contribute to the increase in female-female social interactions following acute isolation, with a particular emphasis on changes in signaling within neural circuits important to the production of USVs [48–50].

Chronic social isolation in rodents leads to increases in anti-social behaviors, including increasing aggressive behavior [4, 7, 10, 11, 13, 14], and studies performed with NIH Swiss and Swiss-Webster male mice also reported that acute social isolation increases aggressive behavior [51, 52]. In contrast, work in male rats found that social isolation for up to 7 days does not increase aggressive behavior [23]. In the current study, we observed fights between male mice only infrequently, and we didn't find any effects of acute social isolation on the occurrence of fighting between pairs of males. Taken together, these findings support the idea that there may be both species and strain differences in the duration of social isolation that is required before the emergence of increased aggression and other anti-social behaviors is observed.

While characterizing the effects of acute isolation on social behavior, we discovered that the relationship of USV production to non-vocal social behaviors differs according to social context. During female-female and male-female encounters, USV production is well related to the amount of time spent the pairs of mice spent interacting, and this was true in pairs with group-housed residents as well as pairs with single-housed residents. We propose that in these social contexts, USV production can be used as a proxy for social and sexual motivation, respectively. In contrast, USV production wasn't well-coupled to the total time spent interacting in male-male trials with group-housed residents and we observed low rates of USV production in these trials, indicating that these pairs interacted without producing USVs. Interestingly, this relationship changes following acute isolation, and USV production scales with the total time spent interacting in male-male trials with single-housed residents. A prior study using chronic social isolation also found that USV production was positively related to the total amount of time spent interacting in pairs of males with a single-housed resident but not in pairs with a group-housed resident [15]. The effect of acute isolation on the relationship between USV production and male-male social interaction is intriguing, and additional work is required to elucidate the significance of this change, and more broadly, the role of USV production during male-male social interactions.

One limitation of our approach is that we don't know which mouse is producing USVs during same-sex encounters. Indeed, the difficulty in ascertaining which mouse in a same-sex pair is vocalizing at a given moment has been a major roadblock to studying the behavioral functions of USV production during same-sex encounters. Given that acute isolation increased female resident-initiated social interactions without affecting visitor-initiated social interactions, an attractive possibility is that an increase in USV production by the resident female following acute isolation drives the observed increase in USV rates in female-female pairs. Consistent with this idea, a previous study showed that resident female mice will vocalize to an anesthetized female visitor but not vice versa [53]. Recent studies employing microphone array recordings found that male mice produce most of the USVs in opposite-sex encounters [28, 29] and that both mice in a pair vocalize during male and female same-sex interactions, at least when these interactions occur in a novel environment between previously single-housed animals [26, 27]. Moreover, it was demonstrated that even within a given social interaction, individuals engaged in different behaviors emit different acoustic types of USVs (chasing versus fleeing) [26]. However, microphone array recordings are challenging to implement in

acoustically noisy home cage recordings, and the location of a behavioral encounter (home cage vs. novel chamber) can strongly impact the dynamics of a social interaction [54–56]. For these reasons, we didn't implement microphone array recordings in the current study, and because we don't know which mouse in a pair emitted any given USV, we also did not analyze or compare the acoustic features of USVs produced during different non-vocal behaviors. Whether and how social isolation impacts the acoustic features of USVs and the relationships between vocalization and non-vocal behaviors remain important topics for future study.

What is the communicative significance of the increase in USV production that we observed in female pairs with an acutely isolated resident? More broadly, how does USV production shape mouse social interactions? Mouse pups produce USVs when isolated from the nest, and these calls elicit retrieval by the dam and thus clearly serve as signals for communication [57–59]. In contrast, the communicative role served by USV production in adult mice, and in adult female mice in particular, remains less clear. Both male and female mice accelerate in response to female-emitted USVs in certain contexts [27], indicating that female USVs can influence the behavior of nearby mice. Aside from these short-term effects on the behavior of other mice, USVs produced during same-sex interactions between females have been speculated to reflect social recognition and memory [60, 61] or to play a role in establishing territorial boundaries or social hierarchy [20, 62], although a causal role for USV production in shaping social dominance remains to be demonstrated. At present, it remains unknown whether and how the increased vocal behavior that we observed in female pairs with single-housed residents impacts communication. In future studies, we plan to implement real-time measurements and manipulations of activity in midbrain neurons important to USV production [48] to measure how manipulations of USV production affect social behavior in the short-term and social success in the long-term, in female-female interactions as well as in other social contexts.

## Supporting information

**S1 File.**
(PDF)

## Acknowledgments

Thanks to Tatyana Matveeva and David Smith for reading draft versions of this manuscript. Thanks also to Frank Drake and other CARE staff for their excellent mouse husbandry.

## Author Contributions

**Conceptualization:** Xin Zhao, Patryk Ziobro, Nicole M. Pranic, Katherine A. Tschida.

**Formal analysis:** Samantha Chu, Samantha Rabinovich, William Chan, Jennifer Zhao, Caroline Kornbrek, Zichen He.

**Funding acquisition:** Katherine A. Tschida.

**Investigation:** Xin Zhao, Patryk Ziobro, Nicole M. Pranic, Katherine A. Tschida.

**Methodology:** Xin Zhao, Patryk Ziobro, Nicole M. Pranic, Katherine A. Tschida.

**Supervision:** Katherine A. Tschida.

**Visualization:** Katherine A. Tschida.

**Writing – original draft:** Katherine A. Tschida.

**Writing – review & editing:** Xin Zhao, Patryk Ziobro, Nicole M. Pranic, Samantha Chu, Samantha Rabinovich, William Chan, Jennifer Zhao, Caroline Kornbrek, Zichen He, Katherine A. Tschida.

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
