## [Decision Letter · Decision Letter 0]

11 Jun 2021

PONE-D-21-12891

Sex- and context-dependent effects of acute isolation on vocal and non-vocal social behaviors in mice

PLOS ONE

Dear Dr. Tschida,

Thank you for submitting your manuscript to PLOS ONE. After careful consideration, we feel that it has merit but does not fully meet PLOS ONE’s publication criteria as it currently stands. Therefore, we invite you to submit a revised version of the manuscript that addresses the points raised during the review process.

As the reviewers indicate, the study has merit but needs to be contextualized and explained in greater detail.  For example, the rationale for studying mice needs to be clarified, and methodology needs to be described early in the manuscript to make it easier for readers to understand the data.  In addition, the results should be discussed with reference to existing literature about mouse communication and general frameworks of animal communication.  

We look forward to receiving your revised manuscript.

Kind regards,

Jon Sakata, PhD

Academic Editor

PLOS ONE

Journal Requirements:

Reviewers' comments:

Reviewer's Responses to Questions

**Comments to the Author**

1. Is the manuscript technically sound, and do the data support the conclusions?

Reviewer #1: Partly

Reviewer #2: Yes

2. Has the statistical analysis been performed appropriately and rigorously? 

Reviewer #1: I Don't Know

Reviewer #2: Yes

3. Have the authors made all data underlying the findings in their manuscript fully available?

Reviewer #1: Yes

Reviewer #2: Yes

4. Is the manuscript presented in an intelligible fashion and written in standard English?

Reviewer #1: Yes

Reviewer #2: Yes

5. Review Comments to the Author

Reviewer #1: The methods as they are currently written make interpreting the findings challenging. First, it is not completely clear if different subjects were used in each treatment of if this was a repeated measures design. Second, it is not clear if each focal animal was unfamiliar with its visitor as not all mice were naïve. Third, the statistical approaches are not explained clearly in the main text and refer to parametric tests when non-parametric tests were reported.

Reviewer #2: The manuscript entitled: ‘Sex-and context-dependent effects of acute isolation on vocal and non-vocal social behaviors in mice’ by Zhao et al. is a novel and timely comparison to the literature on the effects of long-term and juvenile isolation on vocal and nonvocal social behavior. The focus across different behavioral contexts, on the relationship between vocal and nonvocal behaviors, on female same-sex interactions, and on the element of time are all quite interesting. A few concerns include the different experiences of groups of mice outside the context of the study and the lack of USV categorization.

Comments:

A hallmark of mouse USVs is that they occur in distinct structural categories that have been associated with different nonvocal behaviors, including some of the behaviors measured in the current study. Did the authors consider categorizing USVs? If the authors did not categorize USVs, they should include a section of the discussion noting the possibility that different USVs could have been associated with different behaviors that they measured.

Since mice were housed in same-sex pairs, both isolated and group-housed males have a potentially important similarity that could influence their USV production and nonvocal behavior- the lack of experience with females. Could the authors comment on this similarity somewhere in the manuscript? Could this similarity potentially counteract some differences between isolated and socially housed males?

The methods points out that some mice had been used in previous experiments, and so had different kinds of social or nonsocial experience. This could potentially create a difference between these groups, particularly because previous experience of males with females can alter male USV production. Did the mice that had been previously used versus not previously used show any differences in the patterns of their calls or other behaviors? Were mice used in the previous study counterbalanced between isolated and social groups in the current study? This information should be included in the manuscript.

Looking at the supplemental data, was there a difference in the distribution of calls over the 30-minute period for pairs of socially housed mice versus pairs containing a singly housed mouse?

The authors note that social isolation may contribute to increased anxiety behavior. One question is therefore whether the effects of social behavior observed in the study are due to specific effects on social behavior, or to general effects on overall behavioral arousal. What do the authors think about this? Would there be a way to distinguish these possibilities?

Minor comments:

In the Discussion, the authors state that mounting in male-male pairs can be considered aggressive behavior, but it is not considered as aggression in their analysis. Is there a reason for this difference?

The authors note that most USVs in male-female interactions are produced by males, but that some are still produced by females. The statement in lines 102-103 should use the same language as in their other comparisons in stating that calls come from a particular pair type rather than a specific sex.

6. PLOS authors have the option to publish the peer review history of their article (what does this mean?). If published, this will include your full peer review and any attached files.

Reviewer #1: No

Reviewer #2: No

---

## [Author Response · Author response to Decision Letter 0]

27 Jun 2021

Detailed responses to the Editor’s comments

1. “… the rationale for studying mice needs to be clarified…”

Our rationale for studying mice is two-fold. First, from a behavioral standpoint, rodents (including mice and rats) respond to social isolation in ways that resemble human responses to social isolation. Namely, they are inherently social animals that seek out social contact following acute isolation (Niesink and van Ree, 1982; Panksepp and Beatty, 1980; Matthews et al., 2016), and exhibit increases in anti-social behavior following long periods of isolation (Wiberg & Grice, 1963, Valzelli, 1973; Zelikowsky et al., 2018, among others). Second, we plan to follow up on the current study by investigating the neural and molecular mechanisms of the effects of isolation on USV production and social behavior. Mice offer an unparalleled number of viral-genetic tools to achieve this goal. The Introduction has language addressing the first point of our rationale (lines 31-35), and we have now also included a brief overview of the second part of our rationale for using mice (lines 47-50).

2. “… methodology needs to be described early in the manuscript to make it easier for readers to understand the data”

We have changed the format of our manuscript to match the PLOS ONE style template, including moving the Materials and Methods section before the Results. We have also added additional methodological details as requested by Reviewer 1 (please see below for detailed responses to Reviewer comments).

3. “In addition, the results should be discussed with reference to existing literature about mouse communication and general frameworks of animal communication”.

Thank you for this suggestion. We have expanded our discussion of prior studies characterizing the relationship between USV acoustic features and non-vocal behaviors as suggested by Reviewer 2 (lines 497-506). We have also added a final paragraph to the Discussion briefly summarizing what is known regarding the role of USV production in mouse communication and discussing our findings with reference to this work (lines 507-524).

Detailed responses to Reviewer feedback 

Responses to Reviewer 1

1. “First, it is not completely clear if different subjects were used in each treatment of if this was a repeated measures design.”

We did not employ a repeated measures design in this study, and each resident animal was only included in a single treatment group. We have added additional language to the Materials and Methods (lines 75-76) to clarify this point.

2. “Second, it is not clear if each focal animal was unfamiliar with its visitor as not all mice were naïve.”

All visitor mice were unfamiliar to the resident animals, and we have added language to the Materials and Methods to clarify this point (line 86). 

3. “Third, the statistical approaches are not explained clearly in the main text and refer to parametric tests when non-parametric tests were reported.”

We apologize for the confusion. The Reviewer is correct that although our Methods section emphasized parametric statistical tests as the default, we primarily employed non-parametric statistics. To determine whether to use parametric vs. non-parametric tests for a given comparison, we first calculated and plotted the residuals of the relevant data distributions to determine whether these residuals were normally distributed (as determined by visual inspection of plots of z-scored residuals). For example, in Figure 1A, we quantify the effects of isolation on USV rates in the three social contexts. We found that the residuals of the USV rates recorded during male-male social interactions were not normally distributed and therefore employed a non-parametric statistical test (Mann Whitney U test). For the sake of consistency and to be conservative in our conclusions, we then by default applied the same non-parametric statistical test to all other parallel comparisons (i.e., the same comparison made for female-female interactions and for male-female interactions in Figure 1A). For both parametric and non-parametric comparisons, we employed two-side statistical tests with alpha=0.05.

A detailed description of the statistical tests employed is included in S1 Table (included in our initial submission), and we have added language to the “Quantification and statistical analyses” section within the Materials and Methods to clarify our selection and use of statistical analyses (lines 123-131).

Responses to Reviewer 2

1. “Did the authors consider categorizing USVs? If the authors did not categorize USVs, they should include a section of the discussion noting the possibility that different USVs could have been associated with different behaviors that they measured.”

We completely agree with the Reviewer that USVs with different acoustic features may be associated with different behaviors that we measured in the present study. However, because we were not able to assign USVs to specific individuals within a given pair of mice, it is unclear whether a given USV should be compared to the behavior at that moment of the resident mouse or to the behavior of the visitor. At present, the question of whether social isolation impacts the acoustic features of USVs and/or the relationship between USV acoustic features and non-vocal behaviors remains an important topic that we plan to address in future work. We have followed the Reviewer’s recommendation to add language to the Discussion noting the possibility that different types of USVs are associated with different non-vocal behaviors in our dataset and also emphasizing previous studies that have found such a relationship (lines 497-506). 

2. “Since mice were housed in same-sex pairs, both isolated and group-housed males have a potentially important similarity that could influence their USV production and nonvocal behavior- the lack of experience with females. Could the authors comment on this similarity somewhere in the manuscript? Could this similarity potentially counteract some differences between isolated and socially housed males?”

This is a very interesting point! We have now clarified in the Materials and Methods that although a subset of the resident animals used in the male-male recordings and female-female recordings had prior (brief) social experiences with females, all of the resident males used in our male-female recordings were sexually naïve (lines 100-102). It would be an interesting future direction to examine whether and how previous sexual experience influences the effects of social isolation on vocal and non-vocal behaviors, and we now mention this possibility in the Discussion (lines 416-418). 

3. “Did the mice that had been previously used versus not previously used show any differences in the patterns of their calls or other behaviors? Were mice used in the previous study counterbalanced between isolated and social groups in the current study? This information should be included in the manuscript.”

In the current study, a subset of resident mice in the male-male and female-female interactions had a brief amount of prior social experience (no greater than 40 minutes in total), and these mice with previous brief social experiences were counterbalanced between the single-housed and group-housed groups. None of the male residents used in our male-female social interactions had prior social experience. We have added language to the Materials and Methods to clarify these important points (lines 95-102). There were no significant differences in USV rates between socially naïve mice and mice that had received brief, prior social experience (p=0.08 for difference between naïve and experienced in female-female trials, p=0.28 for difference in male-male trials; Mann Whitney test). These statistics have also been added to the Materials and Methods (lines 98-100).

4. “Looking at the supplemental data, was there a difference in the distribution of calls over the 30-minute period for pairs of socially housed mice versus pairs containing a singly housed mouse?”

The Reviewer raises a useful point that although the total numbers of USV produced over 30 minutes may not differ in some social contexts between pairs with a group-housed resident vs. single-housed resident, there may be differences in the temporal distribution of calls within the 30 minutes. Although there is plenty of trial-to-trial variability in USV rates over time, USV production peaks in the first 5 minutes in many trials, particularly in those with single-housed residents. To compare the temporal dynamics of USV production in pairs of group-housed mice versus pairs with a single-housed resident, we have calculated the total number of USVs produced by each pair in the first 5 minutes of the test and divided that value by the total number of USVs recorded in the entire 30 minute trial. In female-female trials, these values are significantly higher in pairs with a group-housed resident (p=0.02, Mann Whitney U test), which reflects the fact these pairs tend to produce USVs in the first 5 minutes and less frequently later on in the trial compared to pairs with a single-housed female. In both male-male and male-female interactions, the proportion of total USVs produced in the first 5 minutes was significantly greater in pairs with single-housed residents (p=0.04 for difference in male-female trials, p=0.01 in male-male trials, Mann Whitney U tests). We have included these analyses in the Results section (lines 171-175, 204-209, and 211-215).

5. “One question is therefore whether the effects of social behavior observed in the study are due to specific effects on social behavior, or to general effects on overall behavioral arousal. What do the authors think about this? Would there be a way to distinguish these possibilities?”

Thank you for raising this interesting question! We agree with the Reviewer that it is entirely possible (and likely) that the effects of acute social isolation are not restricted to or necessarily specific to social motivation but could include effects on behavioral arousal, anxiety, etc. We are in the process of addressing this question more extensively for another study aimed at understanding the neural circuit basis for changes in social and also potentially non-social behaviors following acute isolation in female-female pairs. We plan to perform open field tests to examine locomotion and exploration and elevated plus maze tests to measure anxiety. We have included some pilot data below for the Reviewer’s reference, which show a trend toward decreased movement and exploration of a novel chamber in acutely single-housed versus group-housed females (N=8 group-housed females, N=7 single-housed females, p=0.04, Mann Whitney U test). Because collection of these follow-up behavioral data is in progress for inclusion in a different study, we have not included them in the revised manuscript.

6. “In the Discussion, the authors state that mounting in male-male pairs can be considered aggressive behavior, but it is not considered as aggression in their analysis. Is there a reason for this difference?”

It is true that studies typically consider male-male mounting to be a low-level aggressive behavior. Nonetheless, mounting is a very distinct behavior from attacking, and for that reason, we considered it separately in our analyses. We have also clarified in the Materials and Methods that fighting was defined as attacking and/or biting (line 110).

7. “The authors note that most USVs in male-female interactions are produced by males, but that some are still produced by females. The statement in lines 102-103 should use the same language as in their other comparisons in stating that calls come from a particular pair type rather than a specific sex.”

Thank you for noting this inconsistency, and we have revised that sentence (now lines 207-209).

---

## [Decision Letter · Decision Letter 1]

21 Jul 2021

Sex- and context-dependent effects of acute isolation on vocal and non-vocal social behaviors in mice

PONE-D-21-12891R1

Dear Dr. Tschida,

We’re pleased to inform you that your manuscript has been judged scientifically suitable for publication and will be formally accepted for publication once it meets all outstanding technical requirements.  Thank you for your patience throughout the process. 

Kind regards,

Jon T Sakata, PhD

Academic Editor

PLOS ONE

Additional Editor Comments (optional):

Reviewers' comments:

Reviewer's Responses to Questions

**Comments to the Author**

1. If the authors have adequately addressed your comments raised in a previous round of review and you feel that this manuscript is now acceptable for publication, you may indicate that here to bypass the “Comments to the Author” section, enter your conflict of interest statement in the “Confidential to Editor” section, and submit your "Accept" recommendation.

Reviewer #2: All comments have been addressed

2. Is the manuscript technically sound, and do the data support the conclusions?

Reviewer #2: Yes

3. Has the statistical analysis been performed appropriately and rigorously? 

Reviewer #2: Yes

4. Have the authors made all data underlying the findings in their manuscript fully available?

Reviewer #2: Yes

5. Is the manuscript presented in an intelligible fashion and written in standard English?

Reviewer #2: Yes

6. Review Comments to the Author

Reviewer #2: (No Response)

7. PLOS authors have the option to publish the peer review history of their article (what does this mean?). If published, this will include your full peer review and any attached files.

Reviewer #2: No

---

## [Editor Report · Acceptance letter]

4 Aug 2021

PONE-D-21-12891R1 

Sex- and context-dependent effects of acute isolation on vocal and non-vocal social behaviors in mice 

Dear Dr. Tschida:

I'm pleased to inform you that your manuscript has been deemed suitable for publication in PLOS ONE. Congratulations! Your manuscript is now with our production department. 

Kind regards, 

on behalf of

Dr. Jon T Sakata 

Academic Editor

PLOS ONE